# The Self-Normalized Estimator for Counterfactual Learning

**Adith Swaminathan**
Department of Computer Science
Cornell University
adith@cs.cornell.edu

**Thorsten Joachims**
Department of Computer Science
Cornell University
tj@cs.cornell.edu

## Abstract

This paper identifies a severe problem of the counterfactual risk estimator typically used in batch learning from logged bandit feedback (BLBF), and proposes the use of an alternative estimator that avoids this problem. In the BLBF setting, the learner does not receive full-information feedback like in supervised learning, but observes feedback only for the actions taken by a historical policy. This makes BLBF algorithms particularly attractive for training online systems (e.g., ad placement, web search, recommendation) using their historical logs. The Counterfactual Risk Minimization (CRM) principle [1] offers a general recipe for designing BLBF algorithms. It requires a counterfactual risk estimator, and virtually all existing works on BLBF have focused on a particular unbiased estimator. We show that this conventional estimator suffers from a *propensity overfitting* problem when used for learning over complex hypothesis spaces. We propose to replace the risk estimator with a self-normalized estimator, showing that it neatly avoids this problem. This naturally gives rise to a new learning algorithm – Normalized Policy Optimizer for Exponential Models (Norm-POEM) – for structured output prediction using linear rules. We evaluate the empirical effectiveness of Norm-POEM on several multi-label classification problems, finding that it consistently outperforms the conventional estimator.

## 1 Introduction

Most interactive systems (e.g. search engines, recommender systems, ad platforms) record large quantities of log data which contain valuable information about the system's performance and user experience. For example, the logs of an ad-placement system record which ad was presented in a given context and whether the user clicked on it. While these logs contain information that should inform the design of future systems, the log entries do not provide supervised training data in the conventional sense. This prevents us from directly employing supervised learning algorithms to improve these systems. In particular, each entry only provides *bandit feedback* since the loss/reward is only observed for the particular action chosen by the system (e.g. the presented ad) but not for all the other actions the system could have taken. Moreover, the log entries are *biased* since actions that are systematically favored by the system will by over-represented in the logs.

Learning from historical logs data can be formalized as *batch learning from logged bandit feedback* (BLBF) [2, 1]. Unlike the well-studied problem of online learning from bandit feedback [3], this setting does not require the learner to have interactive control over the system. Learning in such a setting is closely related to the problem of off-policy evaluation in reinforcement learning [4] – we would like to know how well a new system (policy) would perform *if it had been* used in the past. This motivates the use of counterfactual estimators [5]. Following an approach analogous to Empirical Risk Minimization (ERM), it was shown that such estimators can be used to design learning algorithms for batch learning from logged bandit feedback [6, 5, 1].

However the conventional counterfactual risk estimator used in prior works on BLBF exhibits severe anomalies that can lead to degeneracies when used in ERM. In particular, the estimator exhibits a new form of *Propensity Overfitting* that causes severely biased risk estimates for the ERM minimizer. By introducing multiplicative control variates, we propose to replace this risk estimator with a *Self-Normalized Risk Estimator* that provably avoids these degeneracies. An extensive empirical evaluation confirms that the desirable theoretical properties of the Self-Normalized Risk Estimator translate into improved generalization performance and robustness.

## 2 Related work

Batch learning from logged bandit feedback is an instance of causal inference. Classic inference techniques like propensity score matching [7] are, hence, immediately relevant. BLBF is closely related to the problem of learning under covariate shift (also called domain adaptation or sample bias correction) [8] as well as off-policy evaluation in reinforcement learning [4]. Lower bounds for domain adaptation [8] and impossibility results for off-policy evaluation [9], hence, also apply to propensity score matching [7], costing [10] and other importance sampling approaches to BLBF.

Several counterfactual estimators have been developed for off-policy evaluation [11, 6, 5]. All these estimators are instances of importance sampling for Monte Carlo approximation and can be traced back to What-If simulations [12]. Learning (upper) bounds have been developed recently [13, 1, 14] that show that these estimators can work for BLBF. We additionally show that importance sampling can overfit in hitherto unforeseen ways with the capacity of the hypothesis space during learning. We call this new kind of overfitting *Propensity Overfitting*.

Classic variance reduction techniques for importance sampling are also useful for counterfactual evaluation and learning. For instance, importance weights can be "clipped" [15] to trade-off bias against variance in the estimators [5]. Additive control variates give rise to regression estimators [16] and doubly robust estimators [6]. Our proposal uses *multiplicative* control variates. These are widely used in financial applications (see [17] and references therein) and policy iteration for reinforcement learning (e.g. [18]). In particular, we study the self-normalized estimator [12] which is superior to the vanilla estimator when fluctuations in the weights dominate the variance [19]. We additionally show that the self-normalized estimator neatly addresses propensity overfitting.

## 3 Batch learning from logged bandit feedback

Following [1], we focus on the stochastic, cardinal, contextual bandit setting and recap the essence of the CRM principle. The inputs of a structured prediction problem $x \in \mathcal{X}$ are drawn i.i.d. from a fixed but unknown distribution $\Pr(\mathcal{X})$. The outputs are denoted by $y \in \mathcal{Y}$. The hypothesis space $\mathcal{H}$ contains stochastic hypotheses $h(\mathcal{Y} \mid x)$ that define a probability distribution over $\mathcal{Y}$. A hypothesis $h \in \mathcal{H}$ makes predictions by sampling from the conditional distribution $y \sim h(\mathcal{Y} \mid x)$. This definition of $\mathcal{H}$ also captures deterministic hypotheses. For notational convenience, we denote the probability distribution $h(\mathcal{Y} \mid x)$ by $h(x)$, and the probability assigned by $h(x)$ to $y$ as $h(y \mid x)$. We use $(x, y) \sim h$ to refer to samples of $x \sim \Pr(\mathcal{X}), y \sim h(x)$, and when clear from the context, we will drop $(x, y)$.

Bandit feedback means we only observe the feedback $\delta(x, y)$ for the specific $y$ that was predicted, but not for any of the other possible predictions $\mathcal{Y} \setminus \{y\}$. The feedback is just a number, called the loss $\delta : \mathcal{X} \times \mathcal{Y} \mapsto \mathbb{R}$. Smaller numbers are desirable. In general, the loss is the (noisy) realization of a stochastic random variable. The following exposition can be readily extended to the general case by setting $\delta(x, y) = \mathbb{E}[\delta \mid x, y]$. The expected loss – called risk – of a hypothesis $R(h)$ is

$$R(h) = \mathbb{E}_{x \sim \Pr(\mathcal{X})} \mathbb{E}_{y \sim h(x)} [\delta(x, y)] = \mathbb{E}_h [\delta(x, y)]. \tag{1}$$

The aim of learning is to find a hypothesis $h \in \mathcal{H}$ that has minimum risk.

**Counterfactual estimators.** We wish to use the logs of a historical system to perform learning. To ensure that learning will not be impossible [9], we assume the historical algorithm whose predictions we record in our logged data is a stationary policy $h_0(x)$ with full support over $\mathcal{Y}$. For a new hypothesis $h \neq h_0$, we cannot use the empirical risk estimator used in supervised learning [20] to directly approximate $R(h)$, because the data contains samples drawn from $h_0$ while the risk from Equation (1) requires samples from $h$.

Importance sampling fixes this distribution mismatch,

$$R(h) = \mathbb{E}_h\left[\delta(x,y)\right] = \mathbb{E}_{h_0}\left[\delta(x,y)\frac{h(y\,|\,x)}{h_0(y\,|\,x)}\right].$$

So, with data collected from the historical system

$$\mathcal{D} = \{(x_1, y_1, \delta_1, p_1), \ldots, (x_n, y_n, \delta_n, p_n)\},$$

where $(x_i, y_i) \sim h_0$, $\delta_i \equiv \delta(x_i, y_i)$ and $p_i \equiv h_0(y_i\,|\,x_i)$, we can derive an unbiased estimate of $R(h)$ via Monte Carlo approximation,

$$\hat{R}(h) = \frac{1}{n}\sum_{i=1}^{n}\delta_i\frac{h(y_i\,|\,x_i)}{p_i}. \tag{2}$$

This classic inverse propensity estimator [7] has unbounded variance: $p_i \simeq 0$ in $\mathcal{D}$ can cause $\hat{R}(h)$ to be arbitrarily far away from the true risk $R(h)$. To remedy this problem, several thresholding schemes have been proposed and studied in the literature [15, 8, 5, 11]. The straightforward option is to cap the propensity weights [15, 1], i.e. pick $M > 1$ and set

$$\hat{R}^M(h) = \frac{1}{n}\sum_{i=1}^{n}\delta_i \min\left\{M, \frac{h(y_i\,|\,x_i)}{p_i}\right\}.$$

Smaller values of $M$ reduce the variance of $\hat{R}^M(h)$ but induce a larger bias.

**Counterfactual Risk Minimization.** Importance sampling also introduces variance in $\hat{R}^M(h)$ through the variability of $\frac{h(y_i|x_i)}{p_i}$. This variance can be drastically different for different $h \in \mathcal{H}$. The CRM principle is derived from a generalization error bound that reasons about this variance using an empirical Bernstein argument [1, 13]. Let $\delta(\cdot, \cdot) \in [-1, 0]$ and consider the random variable $u_h = \delta(x, y) \min\left\{M, \frac{h(y|x)}{h_0(y|x)}\right\}$. Note that $\mathcal{D}$ contains $n$ i.i.d. observations $u_h{}^i$.

**Theorem 1.** *Denote the empirical variance of $u_h$ by $\hat{\boldsymbol{Var}}(u_h)$. With probability at least $1-\gamma$ in the random vector $(x_i, y_i) \sim h_0$, for a stochastic hypothesis space $\mathcal{H}$ with capacity $\mathcal{C}(\mathcal{H})$ and $n \geq 16$,*

$$\forall h \in \mathcal{H}: \quad R(h) \leq \hat{R}^M(h) + \sqrt{\frac{18\hat{\boldsymbol{Var}}(u_h)\log(\frac{10\mathcal{C}(\mathcal{H})}{\gamma})}{n}} + M\frac{15\log(\frac{10\mathcal{C}(\mathcal{H})}{\gamma})}{n-1}.$$

*Proof.* Refer Theorem 1 of [1] and the proof of Theorem 6 of [13]. $\square$

Following Structural Risk Minimization [20], this bound motivates the CRM principle for designing algorithms for BLBF. A learning algorithm should jointly optimize the estimate $\hat{R}^M(h)$ as well as its empirical standard deviation, where the latter serves as a *data-dependent regularizer*.

$$\hat{h}^{CRM} = \underset{h\in\mathcal{H}}{\operatorname{argmin}}\left\{\hat{R}^M(h) + \lambda\sqrt{\frac{\hat{\boldsymbol{Var}}(u_h)}{n}}\right\}. \tag{3}$$

$M > 1$ and $\lambda \geq 0$ are regularization hyper-parameters.

## 4  The Propensity Overfitting problem

The CRM objective in Equation (3) penalizes those $h \in \mathcal{H}$ that are "far" from the logging policy $h_0$ (as measured by their empirical variance $\hat{\boldsymbol{Var}}(u_h)$). This can be intuitively understood as a safeguard against overfitting. However, overfitting in BLBF is more nuanced than in conventional supervised learning. In particular, the unbiased risk estimator of Equation (2) has two anomalies. Even if $\delta(\cdot, \cdot) \in [\triangledown, \triangle]$, the value of $\hat{R}(h)$ estimated on a finite sample need not lie in that range. Furthermore, if $\delta(\cdot, \cdot)$ is translated by a constant $\delta(\cdot, \cdot) + C$, $R(h)$ becomes $R(h) + C$ by linearity of expectation – but the unbiased estimator on a finite sample need not equal $\hat{R}(h) + C$. In short, this risk estimator is not *equivariant* [19]. The various thresholding schemes for importance sampling only exacerbate this effect. These anomalies leave us vulnerable to a peculiar kind of overfitting, as we see in the following example.

**Example 1.** *For the input space of integers $\mathcal{X} = \{1..k\}$ and the output space $\mathcal{Y} = \{1..k\}$, define*

$$\delta(x, y) = \begin{cases} -2 & \text{if } y = x \\ -1 & \text{otherwise.} \end{cases}$$

*The hypothesis space $\mathcal{H}$ is the set of all deterministic functions $f : \mathcal{X} \mapsto \mathcal{Y}$.*

$$h_f(y|x) = \begin{cases} 1 & \text{if } f(x) = y \\ 0 & \text{otherwise.} \end{cases}$$

*Data is drawn uniformly, $x \sim \mathcal{U}(\mathcal{X})$ and $h_0(\mathcal{Y}|x) = \mathcal{U}(\mathcal{Y})$ for all $x$. The hypothesis $h^*$ with minimum true risk is $h_f^*$ with $f^*(x) = x$, which has risk $R(h^*) = -2$.*

When drawing a training sample $\mathcal{D} = ((x_1, y_1, \delta_1, p_1), ..., (x_n, y_n, \delta_n, p_n))$, let us first consider the special case where all $x_i$ in the sample are distinct. This is quite likely if $n$ is small relative to $k$. In this case $\mathcal{H}$ contains a hypothesis $h_{overfit}$, which assigns $f(x_i) = y_i$ for all $i$. This hypothesis has the following empirical risk as estimated by Equation (2):

$$\hat{R}(h_{overfit}) = \frac{1}{n}\sum_{i=1}^{n} \delta_i \frac{h_{overfit}(y_i \mid x_i)}{p_i} = \frac{1}{n}\sum_{i=1}^{n} \delta_i \frac{1}{1/k} \leq \frac{1}{n}\sum_{i=1}^{n} -1\frac{1}{1/k} = -k.$$

Clearly this risk estimate shows severe overfitting, since it can be arbitrarily lower than the true risk $R(h^*) = -2$ of the best hypothesis $h^*$ with appropriately chosen $k$ (or, more generally, the choice of $h_0$). This is in stark contrast to overfitting in full-information supervised learning, where at least the overfitted risk is bounded by the lower range of the loss function. Note that the empirical risk $\hat{R}(h^*)$ of $h^*$ concentrates around $-2$. ERM will, hence, almost always select $h_{overfit}$ over $h^*$.

Even if we are not in the special case of having a sample with all distinct $x_i$, this type of overfitting still exists. In particular, if there are only $l$ distinct $x_i$ in $\mathcal{D}$, then there still exists a $h_{overfit}$ with $\hat{R}(h_{overfit}) \leq -k\frac{l}{n}$. Finally, note that this type of overfitting behavior is not an artifact of this example. Section 7 shows that this is ubiquitous in all the datasets we explored.

Maybe this problem could be avoided by transforming the loss? For example, let's translate the loss by adding 2 to $\delta$ so that now all loss values become non-negative. This results in the new loss function $\delta'(x, y)$ taking values 0 and 1. In conventional supervised learning an additive translation of the loss does not change the empirical risk minimizer. Suppose we draw a sample $\mathcal{D}$ in which not all possible values $y$ for $x_i$ are observed for all $x_i$ in the sample (again, such a sample is likely for sufficiently large $k$). Now there are many hypotheses $h_{overfit'}$ that predict one of the unobserved $y$ for each $x_i$, basically avoiding the training data.

$$\hat{R}(h_{overfit'}) = \frac{1}{n}\sum_{i=1}^{n} \delta_i \frac{h_{overfit'}(y_i \mid x_i)}{p_i} = \frac{1}{n}\sum_{i=1}^{n} \delta_i \frac{0}{1/k} = 0.$$

Again we are faced with overfitting, since many overfit hypotheses are indistinguishable from the true risk minimizer $h^*$ with true risk $R(h^*) = 0$ and empirical risk $\hat{R}(h^*) = 0$.

These examples indicate that this overfitting occurs regardless of how the loss is transformed. Intuitively, this type of overfitting occurs since the risk estimate according to Equation (2) can be minimized not only by putting large probability mass $h(y \mid x)$ on the examples with low loss $\delta(x, y)$, but by maximizing (for negative losses) or minimizing (for positive losses) the sum of the weights

$$\hat{S}(h) = \frac{1}{n}\sum_{i=1}^{n} \frac{h(y_i \mid x_i)}{p_i}. \tag{4}$$

For this reason, we call this type of overfitting *Propensity Overfitting*. This is in stark contrast to overfitting in supervised learning, which we call *Loss Overfitting*. Intuitively, Loss Overfitting occurs because the capacity of $\mathcal{H}$ fits spurious patterns of low $\delta(x, y)$ in the data. In Propensity Overfitting, the capacity in $\mathcal{H}$ allows overfitting of the propensity weights $p_i$ – for positive $\delta$, hypotheses that avoid $\mathcal{D}$ are selected; for negative $\delta$, hypotheses that overrepresent $\mathcal{D}$ are selected.

The variance regularization of CRM combats both Loss Overfitting and Propensity Overfitting by optimizing a more informed generalization error bound. However the empirical variance estimate is also affected by Propensity Overfitting – especially for positive losses. Can we avoid Propensity Overfitting more directly?

# 5 Control variates and the Self-Normalized estimator

To avoid Propensity Overfitting, we must first detect when and where it is occurring. For this, we draw on diagnostic tools used in importance sampling. Note that for any $h \in \mathcal{H}$, the sum of propensity weights $\hat{S}(h)$ from Equation (4) always has expected value 1 under the conditions required for the unbiased estimator of Equation (2).

$$\mathbb{E}\left[\hat{S}(h)\right] = \frac{1}{n}\sum_{i=1}^{n}\int \frac{h(y_i \mid x_i)}{h_0(y_i \mid x_i)} h_0(y_i \mid x_i) \Pr(x_i) dy_i dx_i = \frac{1}{n}\sum_{i=1}^{n}\int 1 \Pr(x_i) dx_i = 1. \quad (5)$$

This means that we can identify hypotheses that suffer from Propensity Overfitting based on how far $\hat{S}(h)$ deviates from its expected value of 1. Since $\frac{h(y|x)}{h_0(y|x)}$ is likely correlated with $\delta(x,y)\frac{h(y|x)}{h_0(y|x)}$, a large deviation in $\hat{S}(h)$ suggests a large deviation in $\hat{R}(h)$ and consequently a bad risk estimate.

How can we use the knowledge that $\forall h \in \mathcal{H} : \mathbb{E}\left[\hat{S}(h)\right] = 1$ to avoid degenerate risk estimates in a principled way? While one could use concentration inequalities to explicitly detect and eliminate overfit hypotheses based on $\hat{S}(h)$, we use control variates to derive an improved risk estimator that directly incorporates this knowledge.

**Control variates.** Control variates – random variables whose expectation is known – are a classic tool used to reduce the variance of Monte Carlo approximations [21]. Let $V(X)$ be a control variate with known expectation $\mathbb{E}_X\left[V(X)\right] = v \neq 0$, and let $\mathbb{E}_X\left[W(X)\right]$ be an expectation that we would like to estimate based on independent samples of $X$. Employing $V(X)$ as a multiplicative control variate, we can write $\mathbb{E}_X\left[W(X)\right] = \frac{\mathbb{E}[W(X)]}{\mathbb{E}[V(X)]}v$. This motivates the ratio estimator

$$\hat{W}^{SN} = \frac{\sum_{i=1}^{n} W(X_i)}{\sum_{i=1}^{n} V(X_i)}v, \quad (6)$$

which is called the *Self-Normalized estimator* in the importance sampling literature [12, 22, 23]. This estimator has substantially lower variance if $W(X)$ and $V(X)$ are correlated.

**Self-Normalized risk estimator.** Let us use $S(h)$ as a control variate for $R(h)$, yielding

$$\hat{R}^{SN}(h) = \frac{\sum_{i=1}^{n} \delta_i \frac{h(y_i|x_i)}{p_i}}{\sum_{i=1}^{n} \frac{h(y_i|x_i)}{p_i}}. \quad (7)$$

Hesterberg reports that this estimator tends be more accurate than the unbiased estimator of Equation (2) when fluctuations in the sampling weights dominate the fluctuations in $\delta(x,y)$ [19].

Observe that the estimate is just a convex combination of the $\delta_i$ observed in the sample. If $\delta(\cdot,\cdot)$ is now translated by a constant $\delta(\cdot,\cdot) + C$, both the true risk $R(h)$ and the finite sample estimate $\hat{R}^{SN}(h)$ get shifted by $C$. Hence $\hat{R}^{SN}(h)$ is equivariant, unlike $\hat{R}(h)$ [19]. Moreover, $\hat{R}^{SN}(h)$ is always bounded within the range of $\delta$. So, the overfitted risk due to ERM will now be bounded by the lower range of the loss, analogous to full-information supervised learning.

Finally, while the self-normalized risk estimator is not unbiased ($\mathbb{E}\left[\frac{\hat{R}(h)}{\hat{S}(h)}\right] \neq \frac{R(h)}{\mathbb{E}[\hat{S}(h)]}$ in general), it is strongly consistent and approaches the desired expectation when $n$ is large.

**Theorem 2.** *Let $\mathcal{D}$ be drawn $(x_i, y_i) \overset{i.i.d.}{\sim} h_0$, from a $h_0$ that has full support over $\mathcal{Y}$. Then,*

$$\forall h \in \mathcal{H} : \quad \Pr(\lim_{n\to\infty} \hat{R}^{SN}(h) = R(h)) = 1.$$

*Proof.* The numerator of $\hat{R}^{SN}(h)$ in (7) are *i.i.d.* observations with mean $R(h)$. Strong law of large numbers gives $\Pr(\lim_{n\to\infty} \frac{1}{n}\sum_{i=1}^{n} \delta_i \frac{h(y_i|x_i)}{p_i} = R(h)) = 1$. Similarly, the denominator has *i.i.d.* observations with mean 1. So, the strong law of large numbers implies $\Pr(\lim_{n\to\infty} \frac{1}{n}\sum_{i=1}^{n} \frac{h(y_i|x_i)}{p_i} = 1) = 1$. Hence, $\Pr(\lim_{n\to\infty} \hat{R}^{SN}(h) = R(h)) = 1$. $\qquad\square$

In summary, the self-normalized risk estimator $\hat{R}^{SN}(h)$ in Equation (7) resolves all the problems of the unbiased estimator $\hat{R}(h)$ from Equation (2) identified in Section 4.

# 6 Learning method: Norm-POEM

We now derive a learning algorithm, called Norm-POEM, for structured output prediction. The algorithm is analogous to POEM [1] in its choice of hypothesis space and its application of the CRM principle, but it replaces the conventional estimator (2) with the self-normalized estimator (7).

**Hypothesis space.** Following [1, 24], Norm-POEM learns stochastic linear rules $h_w \in \mathcal{H}_{lin}$ parametrized by $w$ that operate on a $d-$dimensional joint feature map $\phi(x, y)$.

$$h_w(y \mid x) = \exp(w \cdot \phi(x, y))/\mathbb{Z}(x).$$

$\mathbb{Z}(x) = \sum_{y' \in \mathcal{Y}} \exp(w \cdot \phi(x, y'))$ is the partition function.

**Variance estimator.** In order to instantiate the CRM objective from Equation (3), we need an empirical variance estimate $\hat{Var}(\hat{R}^{SN}(h))$ for the self-normalized risk estimator. Following [23, Section 4.3], we use an approximate variance estimate for the ratio estimator of Equation (6). Using the Normal approximation argument [21, Equation 9.9],

$$\hat{Var}(\hat{R}^{SN}(h)) = \frac{\sum_{i=1}^{n}(\delta_i - \hat{R}^{SN}(h))^2 (\frac{h(y_i|x_i)}{p_i})^2}{(\sum_{i=1}^{n} \frac{h(y_i|x_i)}{p_i})^2}. \tag{8}$$

Using the delta method to approximate the variance [22] yields the same formula. To invoke asymptotic normality of the estimator (and indeed, for reliable importance sampling estimates) we require the true variance of the self-normalized estimator $Var(\hat{R}^{SN}(h))$ to exist. We can guarantee this by thresholding the importance weights, analogous to $\hat{R}^M(h)$.

The benefits of the self-normalized estimator come at a computational cost. The risk estimator of POEM had a simpler variance estimate which could be approximated by Taylor expansion and optimized using stochastic gradient descent. The variance of Equation (8) does not admit stochastic optimization. Surprisingly, in our experiments in Section 7 we find that the improved robustness of Norm-POEM permits fast convergence during training even without stochastic optimization.

**Training objective of Norm-POEM.** The objective is now derived by substituting the self-normalized risk estimator of Equation (7) and its sample variance estimate from Equation (8) into the CRM objective (3) for the hypothesis space $\mathcal{H}_{lin}$. By design, $h_w$ lies in the exponential family of distributions. So, the gradient of the resulting objective can be tractably computed whenever the partition functions $\mathbb{Z}(x_i)$ are tractable. Doing so yields a non-convex objective in the parameters $w$ which we optimize using L-BFGS. The choice of L-BFGS for non-convex and non-smooth optimization is well supported [25, 26]. Analogous to POEM, the hyper-parameters $M$ (clipping to prevent unbounded variance) and $\lambda$ (strength of variance regularization) can be calibrated via counterfactual evaluation on a held out validation set. In summary, the per-iteration cost of optimizing the Norm-POEM objective has the same complexity as the per-iteration cost of POEM with L-BFGS. It requires the same set of hyper-parameters. And it can be done tractably whenever the corresponding supervised CRF can be learnt efficiently. Software implementing Norm-POEM is available at http://www.cs.cornell.edu/~adith/POEM.

# 7 Experiments

We will now empirically verify if the self-normalized estimator as used in Norm-POEM can indeed guard against propensity overfitting and attain robust generalization performance. We follow the Supervised $\mapsto$ Bandit methodology [2, 1] to test the limits of counterfactual learning in a well-controlled environment. As in prior work [1], the experiment setup uses supervised datasets for multi-label classification from the LibSVM repository. In these datasets, the inputs $x \in \mathbb{R}^p$. The predictions $y \in \{0, 1\}^q$ are bitvectors indicating the labels assigned to $x$. The datasets have a range of features $p$, labels $q$ and instances $n$:

| Name | $p$(# features) | $q$(# labels) | $n_{train}$ | $n_{test}$ |
|------|-----------------|---------------|-------------|------------|
| Scene | 294 | 6 | 1211 | 1196 |
| Yeast | 103 | 14 | 1500 | 917 |
| TMC | 30438 | 22 | 21519 | 7077 |
| LYRL | 47236 | 4 | 23149 | 781265 |

POEM uses the CRM principle instantiated with the unbiased estimator while Norm-POEM uses the self-normalized estimator. Both use a hypothesis space isomorphic to a Conditional Random Field (CRF) [24]. We therefore report the performance of a full-information CRF (essentially, logistic regression for each of the $q$ labels independently) as a "skyline" for what we can possibly hope to reach by partial-information batch learning from logged bandit feedback. The joint feature map $\phi(x,y) = x \otimes y$ for all approaches. To simulate a bandit feedback dataset $\mathcal{D}$, we use a CRF with default hyper-parameters trained on $5\%$ of the supervised dataset as $h_0$, and replay the training data 4 times and collect sampled labels from $h_0$. This is inspired by the observation that supervised labels are typically hard to collect relative to bandit feedback. The BLBF algorithms only have access to the Hamming loss $\Delta(y^*, y)$ between the supervised label $y^*$ and the sampled label $y$ for input $x$. Generalization performance $\mathcal{R}$ is measured by the expected Hamming loss on the held-out supervised test set. Lower is better. Hyper-parameters $\lambda, M$ were calibrated as recommended and validated on a $25\%$ hold-out of $\mathcal{D}$ – in summary, our experimental setup is identical to POEM [1]. We report performance of BLBF approaches without $l2-$regularization here; we observed Norm-POEM dominated POEM even after $l2-$regularization. Since the choice of optimization method could be a confounder, we use L-BFGS for all methods and experiments.

**What is the generalization performance of Norm-POEM?** The key question is whether the appealing theoretical properties of the self-normalized estimator actually lead to better generalization performance. In Table 1, we report the test set loss for Norm-POEM and POEM averaged over 10 runs. On each run, $h_0$ has varying performance (trained on random $5\%$ subsets) but Norm-POEM consistently beats POEM.

Table 1: Test set Hamming loss averaged over 10 runs. Norm-POEM significantly outperforms POEM on all four datasets (one-tailed paired difference t-test at significance level of 0.05).

| $\mathcal{R}$ | Scene | Yeast | TMC | LYRL |
|---|---|---|---|---|
| $h_0$ | 1.511 | 5.577 | 3.442 | 1.459 |
| POEM | 1.200 | 4.520 | 2.152 | 0.914 |
| Norm-POEM | 1.045 | 3.876 | 2.072 | 0.799 |
| CRF | 0.657 | 2.830 | 1.187 | 0.222 |

The plot below (Figure 1) shows how generalization performance improves with more training data for a single run of the experiment on the Yeast dataset. We achieve this by varying the number of times we replay the training set to collect samples from $h_0$ ($ReplayCount$). Norm-POEM consistently outperforms POEM for all training sample sizes.

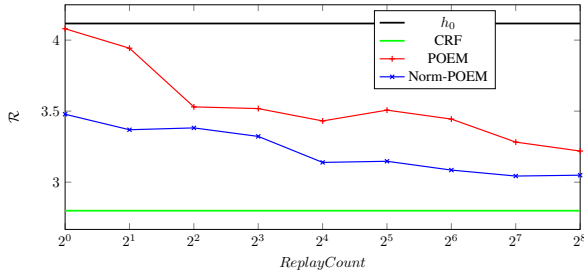

Figure 1: Test set Hamming loss as $n \to \infty$ on the Yeast dataset. All approaches will converge to CRF performance in the limit, but the rate of convergence is slow since $h_0$ is thin-tailed.

**Does Norm-POEM avoid Propensity Overfitting?** While the previous results indicate that Norm-POEM achieves better performance, it remains to be verified that this improved performance is indeed due to improved control over Propensity Overfitting. Table 2 (left) shows the average $\hat{S}(\hat{h})$ for the hypothesis $\hat{h}$ selected by each approach. Indeed, $\hat{S}(\hat{h})$ is close to its known expectation of 1 for Norm-POEM, while it is severely biased for POEM. Furthermore, the value of $\hat{S}(\hat{h})$ depends heavily on how the losses $\delta$ are translated for POEM, as predicted by theory. As anticipated by our earlier observation that the self-normalized estimator is equivariant, Norm-POEM is unaffected by translations of $\delta$. Table 2 (right) shows that the same is true for the prediction error on the test

set. Norm-POEM is consistenly good while POEM fails catastrophically (for instance, on the TMC dataset, POEM is worse than random guessing).

Table 2: Mean of the unclipped weights $\hat{S}(\hat{h})$ (left) and test set Hamming loss $\mathcal{R}$ (right), averaged over 10 runs. $\delta > 0$ and $\delta < 0$ indicate whether the loss was translated to be positive or negative.

|  | $\hat{S}(\hat{h})$ | | | | $\mathcal{R}(\hat{h})$ | | | |
|---|---|---|---|---|---|---|---|---|
|  | Scene | Yeast | TMC | LYRL | Scene | Yeast | TMC | LYRL |
| POEM($\delta > 0$) | 0.274 | 0.028 | 0.000 | 0.175 | 2.059 | 5.441 | 17.305 | 2.399 |
| POEM($\delta < 0$) | 1.782 | 5.352 | 2.802 | 1.230 | 1.200 | 4.520 | 2.152 | 0.914 |
| Norm-POEM($\delta > 0$) | 0.981 | 0.840 | 0.941 | 0.945 | 1.058 | 3.881 | 2.079 | 0.799 |
| Norm-POEM($\delta < 0$) | 0.981 | 0.821 | 0.938 | 0.945 | 1.045 | 3.876 | 2.072 | 0.799 |

**Is CRM variance regularization still necessary?**    It may be possible that the improved self-normalized estimator no longer requires variance regularization. The loss of the unregularized estimator is reported (Norm-IPS) in Table 3. We see that variance regularization still helps.

Table 3: Test set Hamming loss for Norm-POEM and the variance agnostic Norm-IPS averaged over the same 10 runs as Table 1. On Scene, TMC and LYRL, Norm-POEM is significantly better than Norm-IPS (one-tailed paired difference t-test at significance level of 0.05).

| $\mathcal{R}$ | Scene | Yeast | TMC | LYRL |
|---|---|---|---|---|
| Norm-IPS | 1.072 | 3.905 | 3.609 | 0.806 |
| Norm-POEM | 1.045 | 3.876 | 2.072 | 0.799 |

**How computationally efficient is Norm-POEM ?**    The runtime of Norm-POEM is surprisingly faster than POEM. Even though normalization increases the per-iteration computation cost, optimization tends to converge in fewer iterations than for POEM. We find that POEM picks a hypothesis with large $\|w\|$, attempting to assign a probability of 1 to all training points with negative losses. However, Norm-POEM converges to a much shorter $\|w\|$. The loss of an instance *relative* to others in a sample $\mathcal{D}$ governs how Norm-POEM tries to fit to it. This is another nice consequence of the fact that the overfitted risk of $\hat{R}^{SN}(h)$ is bounded and small. Overall, the runtime of Norm-POEM is on the same order of magnitude as those of a full-information CRF, and is competitive with the runtimes reported for POEM with stochastic optimization and early stopping [1], while providing substantially better generalization performance.

Table 4: Time in seconds, averaged across validation runs. CRF is implemented by scikit-learn [27].

| Time(s) | Scene | Yeast | TMC | LYRL |
|---|---|---|---|---|
| POEM | 78.69 | 98.65 | 716.51 | 617.30 |
| Norm-POEM | 7.28 | 10.15 | 227.88 | 142.50 |
| CRF | 4.94 | 3.43 | 89.24 | 72.34 |

We observe the same trends for Norm-POEM when different properties of $h_0$ are varied (e.g. stochasticity and quality), as reported for POEM [1].

# 8   Conclusions

We identify the problem of propensity overfitting when using the conventional unbiased risk estimator for ERM in batch learning from bandit feedback. To remedy this problem, we propose the use of a multiplicative control variate that leads to the self-normalized risk estimator. This provably avoids the anomalies of the conventional estimator. Deriving a new learning algorithm called Norm-POEM based on the CRM principle using the new estimator, we show that the improved estimator leads to significantly improved generalization performance.

**Acknowledgement**

This research was funded in part through NSF Awards IIS-1247637, IIS-1217686, IIS-1513692, the JTCII Cornell-Technion Research Fund, and a gift from Bloomberg.

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
