[Reviews · NeurIPS 2015]

Submitted by Assigned_Reviewer_1

Since I am currently traveling, I only had very little time to finish this (additional) light review. It's unfortunate since I found this paper quite interesting to read. It might very well be that I have missed or misunderstood some points. If this is the case, I apologize.

Observing data from a given hypothesis, we might be interested in evaluating a different hypothesis. If these hypotheses are non-deterministic (e.g. conditional distributions), we can estimate the loss under the hypothesis under investigation by reweighting the data points, see eq. between (1) and (2).

The authors seem to address a problem that is related but somewhat different (see below) from the problem of large variance of the weights. The authors call this problem "propensity overfitting". The examples indicate that this problem occurs if there is not enough exploration, i.e. for a given x, we see only very few or even not a single y that would correspond to the new hypothesis that is to be evaluated. Since a similar argument explains the large variance of the weights, it would be nice to explain the relation between these two problems in a bit more detail.

I didn't have time to look into the details or check novelty of the proposed solution but its idea seems sensible.

- l. 199: I don't get this, if n is small compared to k, we see only few data points with x_i = y_i, right? Then, \hat R(h^) should be even smaller than -2? Am I missing sth.?

- "unbiased counterfactual risk estimator used in prior works on BLBF [4, 5, 1]". I am not sure whether this is a fair statement. I don't know all three papers but I doubt that they are not using sth. like clipping which leads to biased estimation with reduced variance.

- I appreciate that code is available.

Summarizing, I could not check whether the paper is technically sound. But it definitely contains interesting ideas that I would like to think about. I therefore suggest acceptance.
Summary: -

Submitted by Assigned_Reviewer_2

The commentary and notations concerning the main equation - (2) - seem wrong or at least misleading:

an expectation is a quantity of the type \sum f(x) p(x). The expected quantity, f(x), is everything else than p(x), the probability measure.

In equation (2), the loss is so obviously not the expected quantity! Therefore, the following comment (page 2, second paragraph) that (2) has an "anomaly" because it is not invariant to translation of the loss is technically incorrect. Again, the ensuing Example 1 is based on this wrong presumption - that, unsurprisingly, does not hold.

In practical terms, the estimator in (2) is weak to its denominator, p_i, which can be 0 if the sample has never been observed in the prior. The modified estimator in (7) seems better in principle because the effects of p_i are mollified through the double denominator.

I really hope that the authors will be so gracious to modify their commentary should the paper be accepted.

The experimental results show that Norm-POEM outperforms POEM on all datasets and is not far from a CRF using full multi-label information.
Summary: This paper presents an approach for "counterfactual learning", a learning scenario that is a specialisation of multi-label learning. In multi-label learning, the ground-truth annotation for a sample provides a 0/1 label for each of the classes; here, instead, only the label for one class (i.e., bandit feedback).

However, the above is only a justification preamble. De facto, this paper has a precise aim: to improve an estimator (POEM) recently presented by reference [1] at ICML 2015. I would say that, overall, the paper is convincing and worth publication, but with some remarks that I raise in the Comments to authors field.

Submitted by Assigned_Reviewer_3

### Summary

The paper studies the problem of batch learning from logged bandit feedback, a very relevant problem in for example ad ranking. The paper identifies an "overfitting" problem in the recently proposed CRM principle and proposes a solution based on multiplicative control variates (this results in a biased estimator).

### Quality

High-quality and well motivated. Again, as it is follow-up work on [1] I would have expected additional experiments and some additional analysis on the optimization of the training objective (more details than in the paragraph on line 415).

### Clarity

Presentation is clear and paper is well written.

### Originality

Follow-up work on [1], but modifying the risk estimator to be self-normalizing, which seems to make a big difference in the experiments studied by the authors.

### Significance

The problem of learning from logged data is relevant, however it is difficult to say how the substantial experimental improvements translate to more real-world applications.

### Various Remarks

365: if you already run several repetitions, please also include some measure of variance.
Summary: The paper identifies a problem in the recently proposed Conterfactual Risk Minimization (CRM) principle and introduces a solution to the problem. Overall the paper is solid, well written and the work is relevant and novel. I would however have appreciated some extended experiments and especially additional applications then the ones in [1], including some actual real-world applications.

Submitted by Assigned_Reviewer_4

- A question to clarify in the rebuttal: from Section 4.1 in [1], it is said that the CRM approach only makes sense for delta in [-1,0] (and thus they propose to rescale a general loss to [-1,0]). My guess is that it should be the same for NORM-POEM here -- i.e. (8) and (3) both uses the re-scale delta in [-1,0], correct? In this case, I do not understand what it means to use "delta > 0" in the experiment in Table 1. Please clarify.

== Other comments ==

- There is something I find missing in the setup for "batch learning from logged bandit feedback" of Section 3 (or from [1]): I feel the feedback should also be seen as a random variable, rather than a deterministic function as presented on line 110. Here is my rationale. Suppose that the partial feedback is coming from a classification problem where we just do not know the labels, we only get feedback by trying a prediction and seeing what is the loss. Suppose that there is a true classification loss Delta(y',y), and that the way the feedback is generated is, for a given input x, we make a prediction y using h_0; some god agent labels also the input to y'; and then tells us the loss Delta(y',y) that we incur (so here delta(x,y) is Delta(y',y) for the y' given to this example). On the other hand, often in classification, we suppose that the labeling can be noisy and so there is not necessarily a unique y' assigned to each x. This means that somewhere else in the log, we could have the same x as input, the same y that we played, but with a different delta(x,y) as the god agent just had given a different label for this one... This is why I think that in general delta(x,y) should be seen as a random number (in the classification example above the mean of this random variable for a fix x & played y should be the expectation of Delta(y',y) for y' distributed according to p(y'|x), the true noisy labeling distribution).

This perhaps does not change anything about the risk estimator (2); but at least the setup in (1) with the true risk should be presented with an additional expectation over the randomness of delta, to be more general.

- Line 170-171: I think it should be h(y|x) / h_0(y|x). Also, it might be worthwhile to mention that the lack of linearity was already pointed out in [1].

- Line 276: you should probably put argmin instead of argmax given that you are minimizing the risk...

- Line 319: replace bars with hats in the notation to be consistent with the previous convention.

=== Update after rebuttal ==

Thanks for the clarifications.
Summary: [light reviewer]

quality:

6

(out of 10) clarity: 6 originality: 8 significance: 8

I think this is an interesting follow-up work on [1]. I find their analysis of the "propensity overfitting" interesting; and like the idea of using a multiplicative control variate to reduce it. The results show a clear improvement over POEM, on a problem (batch learning from logged bandit feedback) for which there is more and more interest.

Submitted by Assigned_Reviewer_5

The paper propose a method for batch learning from logged bandit feedback that avoids the propensity overfitting problem.

It builds on Swaminathan and Joachims 2015 and provides significantly better algorithm.

Quality: I found the quality of research discussed in the paper to be above average. The extensions proposed in the paper does result in significant improvements in training of the original POEM.

Clarity: For the most part the paper is clearly written and easy to follow.

Originality: The paper gives significantly new formulation to the original POEM algorithm for BLBF problem. I believe the contributions significantly original.

Significance: I think the results in the paper are very interesting and certainly require further research. I am impressed that the proposed algorithm is faster than the original POEM despite having more regularization mechanism built into it. This algorithm may see widespread use in near future.

Summary: This is a good paper. It is clear and reads well.

Submitted by Assigned_Reviewer_6

Counterfactual risk minimization is a technique aimed at learning in a bandit setting but from logged data by correcting for the bias of generating logged examples. While consistent, the paper shows that it suffers from an overfitting problem where a hypothesis class can overfit based on the generation probability of an example. The authors propose to fix this using a multiplicative parameter bounds the variance while maintaining consistency. The experimental results show that the multiplicative (or normalized) counter factual technique outperforms the non-normalized one.

Summary: The paper presents a technique for learning self-normalized estimator for counter-factual learning in the logged bandit setting. It presents ideas for dealing with a form of overfitting where the hypothesis can overfit by having stronger support for the logged data. This is a good paper and should be accepted although I am not absolutely certain.

Author Feedback
Author rebuttal: We thank the reviewers for their valuable comments, which we will gladly implement
to further improve the paper.
Below, we just address comments that may need clarification:

*** Assigned_Reviewer_1:
Re: error bars. A snippet of the 10 runs summarized in Table 1 for the Yeast dataset is attached below.
Run h0 POEM Norm-POEM
1 5.431 4.474 3.386
2 5.778 4.382 4.318
3 5.680 4.575 4.263
4 5.792 4.476 4.470
5 5.644 4.206 4.359
6 5.456 4.656 3.520
7 5.397 4.831 3.215
8 5.716 4.592 4.517
9 5.411 4.458 3.303
10 5.460 4.545 3.405
On each run, Norm-POEM beats POEM which a paired statistical test summarizes accurately.
Much of the variability comes from h0 which is trained on random 5% subsets,
and standard error bars (especially if they overlap) tell a misleading story.

Re: additional analysis of the objective. We will expand on the details in paragraph:321,
deriving how much more the per-iteration cost of Norm-POEM is. We will also qualitatively motivate
why the gradients are better behaved for Norm-POEM.

Re: real-world application. We wished to mimic [1] closely so that, in their controlled experiment setup,
we can clearly see that Norm-POEM dominates POEM. Building on the success of these BLBF approaches,
we have initiated experiments with the internal retrieval systems of Bloomberg and the arXiv search engine.
Preliminary results indicate that bandit feedback (thumbs up, thumbs down for some presented results)
can indeed allow us to learn a high-precision classifier and interleaving feedback
(i.e. clicks on results interleaved with a baseline ranker) can help train a good ranker offline.
We aim to discuss these experiments along with publicly released benchmark BLBF datasets soon.

*** Assigned_Reviewer_4:
Re: notation. The notation is indeed overloaded and unclear, and we will clarify in the final version.
\hat{E}_h0 is supposed to represent the expectation for the empirical distribution
of the sample drawn from h_0 by the Monte Carlo method, which we use as the risk estimator of h.
Essentially, we started with R(h) = \sum delta(y) h(y) where y's were drawn from h,
and we re-wrote it as \sum delta(y) correction_factor h_0(y), now y's are drawn from h_0.
The correction_factor is just a scalar, denoting the ratio of probabilities and as you pointed out,
this is weak to the p_i appearing in the denominator (however p_i can never be 0 since
we would never observe such a sample when y's are drawn from h_0).

*** Assigned_Reviewer_6:
Re: scaling. Norm-POEM is invariant to re-scaling (see line:275 for a quick derivation).
We will highlight this better.

Re: noisy delta. Excellent suggestion, our setup can indeed be motivated with noisy delta,
with a simple additional expectation over delta.

*** Assigned_Reviewer_7:
Re: example. Correct, we intend to say \hat{R}(h*) is around -2 or larger, clearly much larger than -k.

Re: unbiasedness. Correct, related approaches also introduce bias through clipping, we will reword this
as simply, "conventional counterfactual risk estimators".